# Multi-label and Multi-target Sampling of Machine Annotation for Computational Stance Detection

**Zhengyuan Liu[†], Hai Leong Chieu[‡], Nancy F. Chen[†]**
[†]Institute for Infocomm Research (I2R), A*STAR, Singapore
[‡]DSO National Laboratories, Singapore
liu_zhengyuan@i2r.a-star.edu.sg     chaileon@dso.org.sg
nfychen@i2r.a-star.edu.sg

## Abstract

Data collection from manual labeling provides domain-specific and task-aligned supervision for data-driven approaches, and a critical mass of well-annotated resources is required to achieve reasonable performance in natural language processing tasks. However, manual annotations are often challenging to scale up in terms of time and budget, especially when domain knowledge, capturing subtle semantic features, and reasoning steps are needed. In this paper, we investigate the efficacy of leveraging large language models on automated labeling for computational stance detection. We empirically observe that while large language models show strong potential as an alternative to human annotators, their sensitivity to task-specific instructions and their intrinsic biases pose intriguing yet unique challenges in machine annotation. We introduce a multi-label and multi-target sampling strategy to optimize the annotation quality. Experimental results on the benchmark stance detection corpora show that our method can significantly improve performance and learning efficacy.

## 1 Introduction

Stance detection is one of the fundamental social computing tasks in natural language processing, which aims to predict the attitude toward specified targets from a piece of text. It is commonly formulated as a target-based classification problem (Küçük and Can, 2020): given the input text (e.g., online user-generated content) and a specified target, stance detection models are used to predict a categorical label (e.g., *Favor*, *Against*, *None*) (Mohammad et al., 2016; Li et al., 2021). Upon social networking platforms' growing impact on our lives, stance detection is crucial for various downstream tasks such as fact verification and rumor detection, with wide applications including analyzing user feedback and political opinions (Chiluwa and Ifukor, 2015; Ghosh et al., 2019).

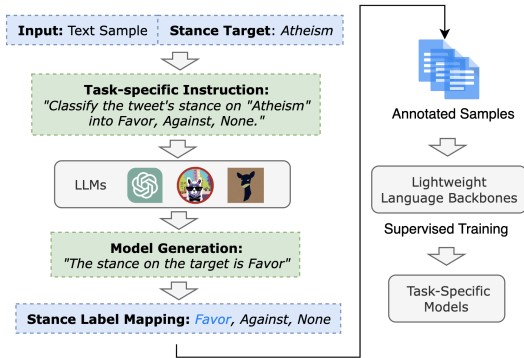

Figure 1: Overview of machine annotation via zero-shot inference with general-purpose large language models.

Data-driven approaches largely benefit from the available human-annotated corpora, which provide domain-specific and task-aligned supervision. Developing state-of-the-art models via in-domain fine-tuning usually requires certain well-annotated training data, such as machine translation and text summarization. However, manual annotations are challenging to scale up in terms of time and budget, especially when domain knowledge, capturing subtle semantic features, and reasoning steps are needed. For instance, for computational stance detection, there are some corpora with target-aware annotation (Mohammad et al., 2016; Allaway and Mckeown, 2020; Li et al., 2021; Glandt et al., 2021), but they are limited in sample size, target diversity, and domain coverage. Learning from insufficient data leads to low generalization on out-of-domain samples and unseen targets (Allaway and Mckeown, 2020), and models are prone to over-fit on superficial and biased features (Kaushal et al., 2021).

Recently, how the versatile large language models (LLMs) would contribute to data annotation raised emerging research interest (He et al., 2023). Machine annotation is also one approach to knowledge distillation from large and general models (Hinton et al., 2015), and student models can be lightweight and cost-effective to deploy. In this paper, we leverage and evaluate LLMs on automated

labeling for computational stance detection. We empirically find that the general-purpose models can provide reasonable stance labeling, and show the potential as an alternative to human annotators. However, there are substantial variances from multiple aspects, and models exhibit biases on target description and label arrangement (§3). Such issues pose challenges regarding the quality and reliability of machine annotation. To optimize the annotation framework, we introduce the multi-label inference and multi-target sampling strategy (§4). Evaluation results on five benchmark stance detection corpora show that our method can significantly improve the performance and learning efficacy (§5).

## 2 Related Work

Computational stance detection aims to automatically predict polarities given the text and a stance target. The model design for this task has been shifted from the traditional statistical methods such as support vector machines (Hasan and Ng, 2013), to neural approaches such as recurrent neural networks (Zarrella and Marsh, 2016), convolutional neural networks (Vijayaraghavan et al., 2016; Augenstein et al., 2016; Du et al., 2017; Zhou et al., 2017), and Transformer (Vaswani et al., 2017). To facilitate the development of data-driven approaches, human-annotated datasets for stance detection are constructed for model training and evaluation (Mohammad et al., 2016; Allaway and Mckeown, 2020; Li et al., 2021; Glandt et al., 2021). While the Transformer-based models, especially pre-trained language backbones (Devlin et al., 2019; Liu et al., 2019), boost the performance on stance detection benchmarks (Ghosh et al., 2019; Li and Caragea, 2021), due to the limited training data and low labeling diversity, these supervised models showed degraded performance on unseen-target and out-of-domain evaluation (Kaushal et al., 2021). To address this problem, recent work introduced de-biasing methods (Clark et al., 2019; Karimi Mahabadi et al., 2020), multi-task learning (Yuan et al., 2022), and the linguistics-inspired augmentation (Liu et al., 2023).

## 3 Machine Annotation for Computational Stance Detection

Based on large-scale pre-training and instruction tuning, language models are capable of solving various downstream tasks in a zero-shot manner (Ouyang et al., 2022). In many NLP benchmarks,

| |
|---|
| **Instruction Prompt A**: Classify a tweet stance on {target-placeholder} into "Favor", "Against", or "None". **Instruction Prompt B**: What is the following tweet's stance on {target-placeholder}? Select one label from "Favor", "Against", or "None". **Instruction Prompt C**: Given the following tweet, give me its stance label on {target-placeholder} from "Favor", "Against", or "None". |
| **Specified Target A**: "atheism" **Specified Target B**: "no belief in gods" **Specified Target C**: "religion" (w/ reversed label) **Specified Target D**: "believe in god" (w/ reversed label) |
| **Label Arrangement A**: Favor, Against, None **Label Arrangement B**: Against, Favor, None **Label Arrangement C**: None, Favor, Against |

Table 1: Different prompt settings for the machine annotation (in the Vicuna style). For the variation of target description, we select *"atheism"* from the *SemEval-16 Task-6 A* (Mohammad et al., 2016) as one example.

they outperform supervised state-of-the-art models, and even achieve comparable performance to human annotators (Bang et al., 2023; Qin et al., 2023). In this section, we investigate the feasibility and reliability of machine annotation for computational stance detection. To compare automated outputs with human annotation, our experiments are conducted on a well-annotated tweet stance detection corpus: *SemEval-16 Task-6 A* (Mohammad et al., 2016), which serves as a benchmark in many previous works. For extensive comparison, we select three general-purpose LLMs: Alpaca-13B (Taori et al., 2023), Vicuna-13B (Chiang et al., 2023), and GPT-3.5-turbo-0301).

### 3.1 Vanilla Machine Annotation Setup

The overview of the machine annotation process is shown in Figure 1. Since stance detection is commonly formulated as a target-based classification task, the text sample and specified target are two elementary components of model input (Allaway and Mckeown, 2020; Liu et al., 2023). To leverage LLMs for automated stance labeling, we formulate task-specific instructions as the prompt (e.g., *"Classify the tweet's stance on the target into Favor, Against, None."*), and adjust their format accordingly to different prompt styles (see details in Appendix Table 4). Since generative system outputs are not in a pre-defined label space, we adopt keyword matching and label re-mapping, and obtain the final model prediction. Moreover, to assess model performance under different settings, we do multiple inference rounds by changing prompts from three aspects, as shown in Table 1.

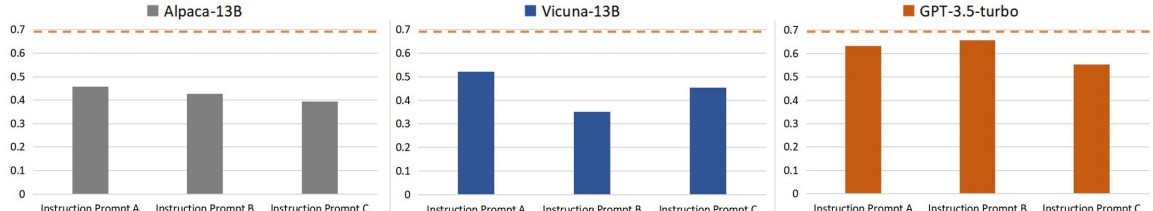

Figure 2: F1 scores of 3-class stance label prediction under 3 different task-specific instructions. The average of human-annotation agreement and supervised training performance is highlighted in the red dotted line.

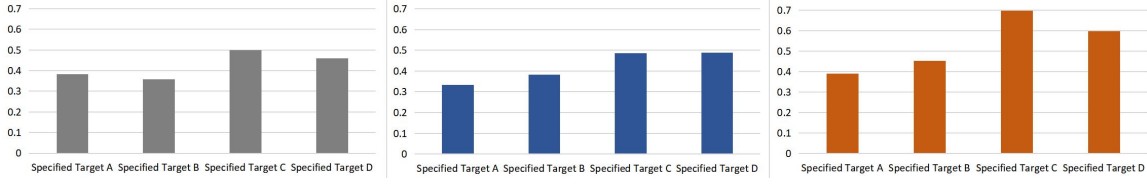

Figure 3: F1 scores of 3-class stance label prediction on 4 different variants of one stance object.

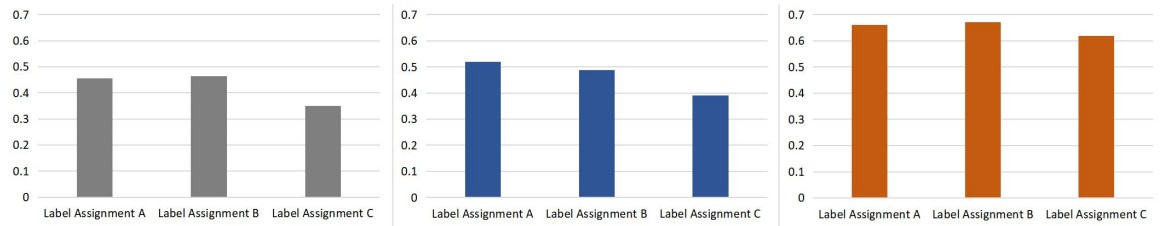

Figure 4: F1 scores of 3-class stance label prediction with 3 different label arrangements.

## 3.2 Vanilla Annotation Result & Analysis

Following previous work (Mohammad et al., 2016; Allaway and Mckeown, 2020), we use the classification metric F1 score to quantitatively measure the model performance. As shown in Figure 2, the general-purpose LLMs can provide reasonable zero-shot inference results, demonstrating the feasibility of machine annotation. In particular, the GPT-3.5-turbo is comparable to human annotators. Not surprisingly, it significantly outperforms Vicuna-13B and Alpaca-13B, since it has a much larger parameter size and is further optimized via human-reinforced instruction tuning. On the other hand, we observe that models do not achieve consistent performance across different prompt settings (see details in Table 1), and the annotation accuracy is prone to fluctuate from multiple aspects:

(1) **Model predictions are sensitive to task-specific instructions.** The nature of human instruction results in diverse prompts for one specific task (Chiang et al., 2023), and there is no gold standard. To simulate this scenario, we paraphrase the instruction for stance detection and keep the prompts semantically identical. We then compare model generations by feeding rephrased instruction prompts. As shown in Figure 2, all models show a certain variance of predicted labels under three different instructions.

(2) **Model predictions are sensitive to the target description.** As stance objects can be in the form of entity, concept, or claim, we then assess the models' robustness on target variants. Here we select *"atheism"* as one example, and use three relevant terms *"no belief in gods"*, *"religion"*, and *"believe in god"* for comparison. In particular, the stance labels of *"atheism"* and *"religion"*/*"believe in god"* are reversed. For instance, if the original label of one text on *"atheism"* is *Favor*, then its label of *"religion"*/*"believe in god"* is *Against*. As shown in Figure 3, models perform much better on the target *"religion"* than the original one, and their results are different across the four variants.

(3) **Model predictions are sensitive to the label arrangement.** Current LLMs still suffer from exposure biases, such as positional bias in summarization, and majority label bias in classification (Fei et al., 2023; Wang et al., 2023). To evaluate the intrinsic bias on label arrangement, we keep the *Instruction Prompt A* fixed, but re-order the stance labels (i.e., *Favor*, *Against*, *None*). As shown in Figure 4, the selected LLMs perform differently given the re-ordered labels, and we observed that prediction of the label *None* is more affected.

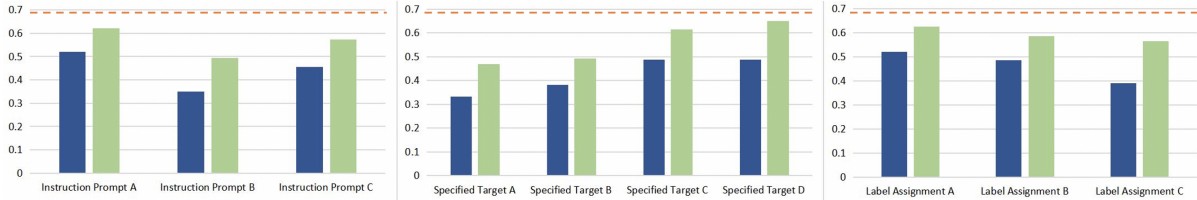

Figure 5: F1 scores of 3-class stance label prediction without (in blue) and with (in green) the two-hop instruction. We evaluate the model `Vicuna-13B` on all three aspects that affect machine annotation performance.

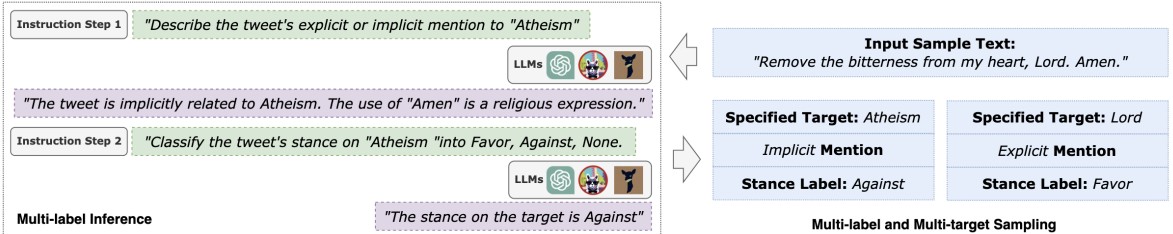

Figure 6: Overview of the multi-label and multi-target machine annotation framework.

## 4 Improving Machine Annotation via Multi-label and Multi-target Sampling

The aforementioned labeling variances pose a question mark on the reliability of machine annotation, and the noisy data cannot be readily used to train dedicated parameter-efficient models. Previous work proved that capturing target referral information and simulating human reasoning steps are beneficial for stance detection (Yuan et al., 2022; Liu et al., 2023). Therefore, we introduce two methods to improve the machine annotation quality. In this section, we evaluate the `Vicuna-13B`, as it is an open model and reaches a balance between performance and computational complexity.

(1) **Enhanced Multi-label Instruction** One challenge of computational stance detection comes from the ubiquitous indirect referral of stance objects (Haddington et al., 2004). Social network users express their subjective attitude with brevity and variety: they often do not directly mention the final target, but mention its related entities, events, or concepts. Inspired by chain-of-thought prompting (Wei et al., 2022) and human behavior in stancetaking (Du Bois, 2007; Liu et al., 2023), to encourage models to explicitly utilize the referral information, we decompose the inference process to two steps. More specifically, as shown in Figure 6, before the stance label prediction, we add one step to describe the context-object relation (i.e., implicit or explicit mention). With the enhanced two-hop instruction, `Vicuna-13B` becomes comparable to the `GPT-3.5-turbo` (see Figure 5).

(2) **Diverse Sampling with Multi-target Prediction** The *SemEval-16 Task-6 A* corpus only covers five stance targets (e.g., *politicians*, *feminism*, *climate change*, *atheism*), and each sample is associated with one target with a human-annotated stance label. To reduce superficial target-related patterns and biases from single target labeling, following previous work (Liu et al., 2023), we apply an adversarial multi-target sampling for machine annotation. First, an off-the-shelf constituency parser is used to extract all noun phrases from the text.[1] We then use these phrases as targets for stance labeling; if one phrase is in a contrary stance label to the original target, we include it as an additional sample. This adversarial multi-target sampling helps models to condition their prediction more on the given target, as well as learn some correlation between explicit and implicit objects. In our experiment, we obtain 1.2k multi-target samples, where the original corpus size is 2914.

## 5 Leveraging Machine-annotated Data for Model Training

Based on our improvements in Section 4, to compare the learning efficacy from machine-annotated and human-annotated data, we conduct experiments by training a supervised model.

**Experimental Setting** As light-weight language backbones are more applicable to practical use cases, we select RoBERTa (Liu et al., 2019) which provides state-of-the-art performance in many pre-

---

[1] https://spacy.io/universe/project/self-attentive-parser

| Model: RoBERTa-base | Original Corpus Training | | | Vanilla Instruction | | |
| Test Set | F1 | Precision | Recall | F1 | Precision | Recall |
|---|---|---|---|---|---|---|
| SemEval-16 Task-6 A | 0.6849 | 0.6755 | 0.7169 | 0.4686 | 0.5496 | 0.4838 |
| SemEval-16 Task-6 B | 0.4134 | 0.5132 | 0.4389 | 0.3969 | 0.5123 | 0.4442 |
| P-Stance | 0.3454 | 0.4840 | 0.3980 | 0.4171 | 0.5155 | 0.4369 |
| VAST | 0.4079 | 0.4215 | 0.4140 | 0.3232 | 0.4286 | 0.4119 |
| Tweet-COVID | 0.3579 | 0.4334 | 0.4032 | 0.3637 | 0.5150 | 0.4088 |

| Model: RoBERTa-base | Enhanced Instruction | | | Multi-target Sampling | | |
| Test Set | F1 | Precision | Recall | F1 | Precision | Recall |
|---|---|---|---|---|---|---|
| SemEval-16 Task-6 A | 0.5514 | 0.5445 | 0.5779 | 0.5553 | 0.5458 | 0.5824 |
| SemEval-16 Task-6 B | 0.4968 | 0.5049 | 0.4975 | 0.5954 | 0.6118 | 0.585 |
| P-Stance | 0.4769 | 0.5125 | 0.4992 | 0.4811 | 0.5116 | 0.4858 |
| VAST | 0.3909 | 0.4362 | 0.4317 | 0.6325 | 0.6443 | 0.6406 |
| Tweet-COVID | 0.5152 | 0.5271 | 0.5178 | 0.5554 | 0.5815 | 0.5638 |

Table 2: Supervised fine-tuning results of the 3-class stance classification. The single-domain training corpus is highlighted in blue, and the model is evaluated on multiple test sets. Macro-averaged F1, Precision, and Recall scores are reported. Training with human-annotated and machine-annotated data is highlighted in green and pink, respectively. Results of 2-class macro-averaged scores are shown in Appendix Table 5.

vious works on stance detection (Liu et al., 2023; Zhao et al., 2023). We train the model on the *SemEval-16 Task-6 A* corpus, and collect five representative benchmarks to build the in-domain, out-of-domain, and cross-target evaluation settings (Küçük and Can, 2020), including *SemEval-16 Task-6 A* and *Task-6 B* (Mohammad et al., 2016), *P-Stance* (Li et al., 2021), *VAST* (Allaway and Mckeown, 2020), and *Tweet-COVID* (Glandt et al., 2021). Model input is formulated as " {target}   {context} ", and the final hidden representation of the first token "" is fed to a linear layer and softmax function to compute the output probabilities. Experimental details are shown in Appendix Table 3 and Table 4. We use the 3-class prediction scheme, as the *None* label is necessary for practical use cases. Following previous work (Mohammad et al., 2016; Allaway and Mckeown, 2020), we adopt the macro-averaged F1, Precision, and Recall as evaluation metrics.

**Results and Analysis** Since we train the model on a single-domain corpus, testing it on out-of-domain samples and various unseen targets poses a challenging task. As shown in Table 2, compared with training on human-annotated data, applying machine annotation upon the vanilla instruction in Section 3 results in lower scores, especially on the *SemEval-16 Task-6 A*; F1, precision, and recall scores are all affected. When applying our proposed methods, we observe that:

(1) **The multi-label instruction can improve in-domain performance.** Our proposed two-hop instruction provides training data with higher quality, particularly in the in-domain evaluation (17.6% rel-

ative gain); this is consistent with the improved annotation accuracy shown in Figure 5.

(2) **The multi-target sampling can improve out-of-domain and cross-target performance.** The single-target data under vanilla and enhanced two-hop instruction only enable the model to achieve reasonable results on in-domain samples and existing targets. In contrast, the multi-target sampling brings substantial and consistent improvement on out-of-domain and unseen targets, where relative gains on *SemEval-16 Task-6B* and *VAST* are 19.8% and 61.2%, and it approaches 83% of supervised training on human-annotated datasets. This demonstrates that the model learns more general and domain-invariant features. We also report 2-class macro-averaged scores (i.e., *Favor*, *Against*), where the *None* label is only discarded during inference (see Appendix Table 5), and our methods bring performance elevation at all fronts.

## 6  Conclusions

In this work, we investigated the potential and challenges of leveraging large language models on automated labeling for computational stance detection. Quantitative analyses demonstrated the feasibility, but their predictions are sensitive to task-specific instructions, and LLMs inevitably present exposure biases on target description and label arrangement. We improved machine annotation by adopting a multi-label and multi-target sampling strategy. Experimental results on several stance detection benchmarks showed the effectiveness of our proposed methods. This work can shed light on further extensions of machine annotation.

## Limitations

All samples used in this work are in English, thus to apply the model to other languages, the result might be limited by the multilingual capability of language backbones. Moreover, we are aware that it remains an open problem to mitigate biases in human stancetaking. Of course, current models and laboratory experiments are always limited in this or similar ways. We do not foresee any unethical uses of our proposed methods or their underlying tools, but hope that it will contribute to reducing incorrect system outputs.

## Ethics and Impact Statement

We acknowledge that all of the co-authors of this work are aware of the provided ACL Code of Ethics and honor the code of conduct. All data used in this work are collected from existing published NLP studies. Following previous work, the annotated corpora are only for academic research purposes and should not be used outside of academic research contexts. Our proposed framework and methodology in general do not create a direct societal consequence and are intended to be used to prevent data-driven models from over-fitting domain-dependent and potentially-biased features.

## Acknowledgments

This research was supported by funding from the Institute for Infocomm Research (I$^2$R), A*STAR, Singapore, and DSO National Laboratories, Singapore. We thank Yong Keong Yap and Wen Haw Chong for the research discussions. We also thank the anonymous reviewers for their precious feedback to help improve and extend this piece of work.

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

| Corpus | Original Targets for Stance Labeling | Train | Valid | Test |
|---|---|---|---|---|
| SemEval-16 Task-6 A | Atheism, Climate Change, Feminist Movement, Hillary Clinton, Legalization of Abortion | 2914 | - | 1249 |
| SemEval-16 Task-6 B | Donald Trump (for zero-shot evaluation) | - | - | 707 |
| P-Stance | Donald Trump, Joe Biden, Bernie Sanders | 19228 | 2462 | 2374 |
| VAST | Various Targets by Human Annotation | 13477 | 2062 | 3006 |
| Tweet-COVID | Keeping Schools Closed, Dr. Fauci, Stay at Home Orders, Wearing a Face Mask | 4533 | 800 | 800 |

Table 3: Statistics of the collected stance detection datasets for model training and evaluation.

| Machine Annotation | Experimental Configuration |
|---|---|
| Alpaca-13B | Below is an instruction that describes a task, paired with an input that provides further context. Write a response that appropriately completes the request.
### Instruction: Select one stance label from "Favor", "Against", or "None", with one short sentence.
### Input: Given the tweet: {tweet-placeholder}, what is its stance on the target {target-placeholder}?
### Response: {model-output} |
| Vicuna-13B | Classify a tweet stance on {target-placeholder} into "Favor", "Against", or "None".
Tweet: {tweet-placeholder}
Stance: {model-output} |
| Vicuna-13B | Target Referral Instruction:
Describe the tweet's explicit or implicit relation to the target {target-placeholder}.
Tweet: {tweet-placeholder}
Answer: {model-output} |
| GPT-3.5-turbo | Given a tweet:{tweet-placeholder}, what is its stance to the target {target-placeholder}? Select one label from "Favor", "Against", and "None", with one short sentence. |

| Environment Details | |
|---|---|
| GPU Model | Single Tesla A100 with 40 GB memory; CUDA version 12.0. |
| Library Version | Pytorch==1.13.1; Transformers==4.28.1. |
| Computational Cost | For LLMs inference (Alpaca-13B and Vicuna-13B), the average evaluation time is 1 hour for one round. For GPT-3.5-turbo, we utilize the OpenAI official API. For supervised training, the average training time is 1.5 hours for one round. Average 3 rounds for each reported result (calculating the mean of the scores). |

| Dataset Processing | Details |
|---|---|
| Corpus | The datasets we used for training and evaluation are from published works (Mohammad et al., 2016; Allaway and Mckeown, 2020; Li et al., 2021; Glandt et al., 2021) with the Creative Commons Attribution 4.0 International license. |
| Pre-Processing | All samples are in English, and only for research use. Upper-case, special tokens, and hashtags are retained. |

| Supervised Training | Experimental Configuration |
|---|---|
| RoBERTa-base | RoBERTa-base (Liu et al., 2019)
Base Model: Transformer (12-layer, 768-hidden, 12-heads, 125M parameters).
Learning Rate: 2e-5, AdamW Optimizer, Linear Scheduler: 0.9. |

Table 4: Details of the experimental environment and the hyper-parameter setting.

| Model: RoBERTa-base | Original Corpus Training | | | Vanilla Instruction | | |
| --- | --- | --- | --- | --- | --- | --- |
| Test Set | F1 | Precision | Recall | F1 | Precision | Recall |
| SemEval-16 Task-6 A | 0.7023 | 0.7047 | 0.7318 | 0.5826 | 0.5376 | 0.6496 |
| SemEval-16 Task-6 B | 0.3143 | 0.5126 | 0.2814 | 0.5224 | 0.4526 | 0.6432 |
| P-Stance | 0.4436 | 0.6597 | 0.5118 | 0.6017 | 0.6436 | 0.6442 |
| VAST | 0.3905 | 0.4192 | 0.3906 | 0.4419 | 0.4185 | 0.5711 |
| Tweet-COVID | 0.2575 | 0.4377 | 0.1966 | 0.3409 | 0.3432 | 0.4788 |
| Model: RoBERTa-base | Enhanced Instruction | | | Multi-target Sampling | | |
| Test Set | F1 | Precision | Recall | F1 | Precision | Recall |
| SemEval-16 Task-6 A | 0.5856 | 0.5935 | 0.6037 | 0.6034 | 0.6071 | 0.5911 |
| SemEval-16 Task-6 B | 0.5318 | 0.5111 | 0.5578 | 0.5905 | 0.6085 | 0.5813 |
| P-Stance | 0.5992 | 0.6747 | 0.5968 | 0.6431 | 0.6765 | 0.6596 |
| VAST | 0.4867 | 0.4312 | 0.5834 | 0.5659 | 0.6168 | 0.5382 |
| Tweet-COVID | 0.4711 | 0.4625 | 0.4978 | 0.4791 | 0.5346 | 0.4460 |

Table 5: Results of the 2-class stance classification on multiple corpora. Macro-averaged F1, Precision, and Recall scores are reported. Results of 3-class macro-averaged scores are shown in Table 2.