# OpenReview forum: "Multi-label and Multi-target Sampling of Machine Annotation for Computational Stance Detection"
_EMNLP/2023/Conference — EMNLP 2023 Findings_

### Official Review · Reviewer_KnpT · 2023-07-24

**Typos Grammar Style And Presentation Improvements:** 1. In descriptions in Fig 2-4, I woul…
**Soundness:** 4

**Excitement:**

4: Strong: This paper deepens the understanding of some phenomenon or lowers the barriers to an existing research direction.

**Paper Topic And Main Contributions:**

This work proposes the usage of LLM as an alternative to human annotations and justifies this by conducting experiments on 3 LLMs. Their task setup is target based classification problem. The authors demonstrate what affects models' predictions. Also, the authors finetune Roberta and evaluate it on 5 benchmark corpora.

**Questions For The Authors:**

1. In line 180 - could you please elaborate on how you create these samples, maybe provide an example.
2.  Regarding Figure 5 and all other measurements, how many runs were done per measurement? do you have confidence intervals for your measurements?


**Reasons To Accept:**

Show how LLMs work for automated labeling, and show what aspects affect their performance. In addition, they show a model's performance when finetuning with machine annotations: they show an improvement in the performance for out-of-domain corpora compared to the model trained on human annotations.

**Reasons To Reject:**

I would hope to get some clarifications for my questions and some of my suggested edits to be implemented. Given that clarifications make sense, I don't have strong reasons for the rejection.

**Reproducibility:**

3: Could reproduce the results with some difficulty. The settings of parameters are underspecified or subjectively determined; the training/evaluation data are not widely available.

**Reviewer Confidence:**

4: Quite sure. I tried to check the important points carefully. It's unlikely, though conceivable, that I missed something that should affect my ratings.

---

> ### Author Rebuttal · Authors · 2023-08-28
>
> #### **1) For the adversarial multi-target labeling process**
> The adversarial multi-target labeling basically means to **select the explicit object that shows a stance dis-alignment to the specified target**. Previous work (Liu et al., 2023) has demonstrated its effectiveness **to reduce superficial target-related patterns and biases from single target labeling** (the issues resulting from low data diversity and spurious correlation [1][2]), and emphasize object-object relationship.
>
> For the detailed adversarial multi-target labeling process, here is one example:
>
> ```
> Given the tweet “So can unborn children have rights now?” with the original specified target “abortion” for stance analysis, 4 steps are used to generate multi-target labeling:
>
> >>> Step 1: The machine annotator predicts a label “AGAINST” of the implicit target “abortion”.
>
> >>> Step 2: We run an off-the-shelf constituency parser to extract the noun phrases (explicit targets) from the tweet. In this example they are “unborn children” and “rights”.
>
> >>> Step 3: For both of the explicit targets “unborn children” and “rights”, the machine annotator predicts the stance label “FAVOR”, which is a contrary label to the original specified target “abortion”. If no such explicit target is found, Step 4 will be skipped.
>
> >>> Step 4: We randomly select one of the explicit targets with the contrary label (e.g., “unborn children”), and then obtain an adversarial multi-target labeling of the tweet (“abortion” -> “AGAINST”, “unborn children” -> “FAVOR”).
>
> ```
> ####
> Aside from the framework overview shown in Figure 6, we will illustrate the above example in appendix.
>
> ####
> #### **2) For the experimental setting of running times**
> The experiments were run three times, and the mean value is reported in this manuscript. These setups are included in the “Environment Details” section in Appendix Table 3 on Page 7. We could include the means ± SDs in the result tables.
> Given the reproducibility, the code and data described in this manuscript will be released in our final version upon acceptance.
>
> #### **3) For the presentation improvements (figures/tables readability)**
> Thanks for your suggestions!
> We will (1) specify the “instruction prompt”, “specified target” and “label assignment” in the corresponding figures; (2) include GPT3.5 performance data in Figure 5 for better comparison; and (3) highlight the best results in Table 1.
>
> ####
> ####
> **We thank the Reviewer KnpT for the positive comments of the novelty and soundness of our work.
> Hopefully we have addressed all the questions. We believe our work can shed light on the future studies of machine annotation.**
>
> ####
> #### **Reference:**
> [1] Shalmoli Ghosh, Prajwal Singhania, Siddharth Singh, Koustav Rudra, and Saptarshi Ghosh. 2019. Stance detection in web and social media: a comparative study. In International Conference of the CrossLanguage Evaluation Forum for European Languages, pages 75–87. Springer.
> [2] Ayush Kaushal, Avirup Saha, and Niloy Ganguly. 2021. twt–wt: A dataset to assert the role of target entities for detecting stance of tweets. In Proceedings of the 2021 Conference of the North American Chapter of the Association for Computational Linguistics: Human Language Technologies, pages 3879–3889.

---

### Official Review · Reviewer_5SF3 · 2023-07-25

**Soundness:** 3

**Excitement:**

2: Mediocre: This paper makes marginal contributions (vs non-contemporaneous work), so I would rather not see it in the conference.

**Paper Topic And Main Contributions:**

This short paper explores the use of language models for labeling stance detection training data. The paper compares different zero-shot labeling approaches (multi-hop, different instructions), as well as automatically augmenting the training data with additional targets (Liu et al., ACL 2023). The authors then evaluate a classifier trained on those weakly labeled data, showing that the techniques improve the cross-target generalization of the classifier.

**Reasons To Accept:**

- Sound experimental setup
- Good reproducibility thanks to the use of an openly available language model. The authors indicate that they are planning to share their code and/or data, but I could not verify this as no supplementary material was shared.


**Reasons To Reject:**

- Lack of focus: I did not have the impression that this short paper makes a focused contribution, but rather it seems to explore several techniques in a relatively superficial manner. Unfortunately, there is little analysis of the different approaches.

**Reproducibility:**

5: Could easily reproduce the results.

**Reviewer Confidence:**

4: Quite sure. I tried to check the important points carefully. It's unlikely, though conceivable, that I missed something that should affect my ratings.

**Typos Grammar Style And Presentation Improvements:**

- I find Figures 2–4 hard to understand. Consider making the figures more self-contained.
- I find it difficult to compare the four approaches in Tables 1 and 4. Consider changing the table layout to make comparison easier, for example by providing averages.

---

> ### Author Rebuttal · Authors · 2023-08-28
>
> #### **1) Our contributions**
>   In this work, **we focus on assessing and improving the quality of machine annotation for computational stance detection**.
>   We first brought up **the inconsistency of machine annotation** based on our experiments (Section 2), which would significantly affect the robustness yet has not drawn enough attention. By changing the ways of instruction, the expression of targets, and the sequence of labels, we found any alterations resulted in prediction variation. This thus motivates us to reduce the prediction variance introduced by machine annotations.
> Therefore, we then adopted multi-label inference and multi-target sampling strategies **to optimize the machine annotation framework** (Section 3).
> Finally, to demonstrate the effectiveness of our strategies and the reliability of the machine-annotated data generated by our methods, we fine-tuned a task-specific model on our annotated data, and evaluated its performance on five stance detection test sets (Section 4). The **results clearly show that the performance of the model trained on our improved machine-annotated data is on par with that trained upon human annotation**, strongly demonstrating the effectiveness of our proposed strategies to improve the quality of machine annotation.
>
> #### **2) For the novelty of our work**
> **The robustness of machine annotation is important yet has not drawn enough attention.** In this work, to strategically improve machine annotation for computational stance detection, our proposed methods are based on our thorough analyses of prediction variations, and we extend theory-inspired practice from human annotation [1][2] to machine annotation.
>
>
> #### **3) For the presentation improvements (figures/tables readability)**
> Thanks for your suggestions!
> We will (1) specify the “instruction prompt”, “specified target”, and “label assignment” in the corresponding figures, and (2) highlight the best results in Tables 1 and 4.
>
> For your quick reference, here are the 3 human-written instruction prompts (Figure 2) following the Vicuna format:
> > Instruction Prompt A:
> Classify a tweet stance on {target-placeholder} into “Favor”, “Against”, or “None”.
> > Instruction Prompt B:
> What is the following tweet’s stance on {target-placeholder}? Select one label from “Favor”, “Against”, or “None”.
> > Instruction Prompt C:
> Given the following tweet, give me its stance label on {target-placeholder} from “Favor”, “Against”, or “None”.
>
> The target examples in Figure 3 (Section 2) are:
> > “atheism” (Specified target A)
> > “no belief in gods” (Specified target B)
> > “religion” (Specified target C)
> > “believe in god” (Specified target D)
>
> The label sequences in our experimental setting (Figure 4) are (the label order is one instance of label assignment):
> > “Favor, Against, None” (Label Assignment A)
> > “Against, Favor, None” (Label Assignment B)
> > “None, Favor, Against” (Label Assignment C)
>
> Noteworthy, the model prediction variance is **not only related to the above specified instructions**, even small changes to the prompt also make a difference (e.g., change “a tweet” to “the tweet”).
>
> #### **4) For the code and data availability**
> We will release our code and data for both the machine annotation and stance detection experiments. As we used the open-source models Alpaca and Vicuna for machine annotation, and Roberta for stance detection, we suppose that interested readers will be able to reproduce our experiments following our paper and code.
>
> ####
> ####
> **We thank the Reviewer 5SF3 for supporting the soundness and reproducibility of our work.
> Hopefully we have addressed all the comments/questions, and the focus/contribution of our work is better delivered/explained.**
>
> ####
> #### **Reference:**
> [1] Ayush Kaushal, Avirup Saha, and Niloy Ganguly. 2021. twt–wt: A dataset to assert the role of target entities for detecting stance of tweets. In Proceedings of the 2021 Conference of the North American Chapter of the Association for Computational Linguistics: Human Language Technologies, pages 3879–3889.
> [2] Zhengyuan Liu, Yong Keong Yap, Hai Leong Chieu, and Nancy Chen. 2023. Guiding Computational Stance Detection with Expanded Stance Triangle Framework. In Proceedings of the 61st Annual Meeting of the Association for Computational Linguistics, pages 3987–4001, Association for Computational Linguistics.

---

### Official Review · Reviewer_jYAZ · 2023-08-03

**Soundness:** 3

**Excitement:**

2: Mediocre: This paper makes marginal contributions (vs non-contemporaneous work), so I would rather not see it in the conference.

**Missing References:**

The "enhanced" instruction approach looks a lot like chain of thought prompting, exemplified by this paper:  https://arxiv.org/abs/2201.11903  ... it would be appropriate to discuss the connection to those kinds of approaches or to prompt engineering.

**Paper Topic And Main Contributions:**

The paper is about stance detection.  There are two different approaches discussed.  The first explores zero-shot stance detection with a range of different models, showing first that diffferent instruction wording, different target entities (i.e., toward who/what is the author's stance to be determined?), and the ordering of the possible labels (in the prompt) can all have a notable effect on the accuracy.  A "two hop" approach to prompting is introduced and shown to improve performance of one of the language models (vicuna) across these settings.  The second approach turns to finetuning the Roberta model for the task, varying the instructions and substituting automatically generated data for the original training data.

**Reasons To Accept:**

While I appreciate the paper's focus on one worthwhile problem, there was unfortunately very little about the proposed approach that was clearly motivated by, or obviously unique to, the stance detection task.

**Reasons To Reject:**

The experiments in section 2 are extremely unclear.  In figures 2-4 the independent variables are never fully defined, nor are their values (the figure unhelpfully labels them "instruction prompt A," "specified target B," and so on -- the labels are never aligned to an explanation of the design choice).  The text claims, for example, that "mdoels perform much better on the target religion than the original one" but it's not clear which target among A-D is "religion" vs "the original one," or what the original one was.  All the reader is told about the different instructions is "we adjust task-specific instructions according to their format (see appendix table 3)."  I really couldn't make any sense out of what the experiments show, because the explanation was so vague.

Benefits to "learning efficiency" are claimed in the abstract and intro but never discussed again.

A major limitation of this work is its partial dependence on closed models.  The authors/reviewers have no way to verify that the evaluation data was excluded from the pretraining data.

I couldn't follow what was being presented in the last paragraph of section 3.  At first I thought this was going to be another way to improve the zero-shot approach.  But it sounds more like a set of transformations on the original data (i.e., some kind of data augmentation technique).  I couldn't understand what the authors were trying to do, and the claim that it "helps models to condition their prediction more on the given target as well as learn some correlation between explicit and implicit objects" isn't connected, as far as I can tell, to any experiments.

Section 4 is entirely disconnected from the rest of the paper's exposition; it turns to finetuning Roberta (a completely different type of model) to carry out the same task, but now evaluating on a wider range of datasets.  What I take away from the experiments here (table 1) is that training on silver data (I think, the labels produced by the models in section 3?  this is not made perfectly clear) is worse than training on human annotations, but the gap narrows if the proposed techniques are used.

A major gap in the experimental story:  what about combining the gold data with the silver data?  That's what's usually done when models are used to automatically produce training data for supervised learners, and it's more relevant to real world practice.

**Reproducibility:**

3: Could reproduce the results with some difficulty. The settings of parameters are underspecified or subjectively determined; the training/evaluation data are not widely available.

**Reviewer Confidence:**

4: Quite sure. I tried to check the important points carefully. It's unlikely, though conceivable, that I missed something that should affect my ratings.

**Typos Grammar Style And Presentation Improvements:**

Line 133:  "sensitive to the label assignment" -- this is confusing, it would be better to call this "the label order in the prompt"

The results in table 1 make it very hard to compare apples to apples and draw conclusions.  Perhaps a collection of scatterplots (one per test set), with recall and precision as the axes, and different colors showing alternate training set construction methods, would make this easier to understand?

---

> ### Author Rebuttal · Authors · 2023-08-28
>
> #### **1.1) Our contributions**
> In this work, we focus on evaluating and improving the quality of machine annotation for computational stance detection.
> We first **discussed the inconsistency of machine annotation from three aspects  (Section 2)**: the ways of instruction, the expression of targets, and the sequence of labels. Alterations in any of these aspects result in prediction variation. This motivates us to reduce the prediction inconsistency observed in machine annotation.
> Therefore, we then adopted the multi-label inference and the multi-target sampling strategy **to optimize the annotation framework (Section 3)**.
> Finally, to demonstrate the effectiveness of our strategies and the reliability of the machine-annotated data generated by our methods, we **fine-tuned a task-specific model on machine-annotated data, and evaluated its performance on five stance detection test sets (Section 4)**. The results clearly show that the performance of the model trained on our improved machine-annotated data is on par with that trained upon human annotation, strongly demonstrating the effectiveness of our proposed strategies to improve the quality of machine annotation.
> We’re not doing/comparing two approaches as the reviewer commented, i.e., zero-shot stance detection and fine-tuning.
>
> #### **1.2) For the figures shown in Section 2**
> Basically, Figure 2, Figure 3, and Figure 4 are to show the inconsistency in machine annotation. The substantial variation showed up just by changing the ways of instruction (Figure 2), expression of targets (Figure 3), and the sequence of labels (Figure 4), which is surprising and not reported before. We apologize for the confusion, but we believe **the patterns shown in the figures clearly demonstrate our motivation to improve the quality of machine annotation.**
>
> More specifically, for your quick reference, here are the 3 human-written instruction prompts (Figure 2) following the Vicuna format (one is shown in Appendix Table 3):
> > Instruction Prompt A:
> Classify a tweet stance on {target-placeholder} into “Favor”, “Against”, or “None”.
> > Instruction Prompt B:
> What is the following tweet’s stance on {target-placeholder}? Select one label from “Favor”, “Against”, or “None”.
> > Instruction Prompt C:
> Given the following tweet, give me its stance label on {target-placeholder} from “Favor”, “Against”, or “None”.
>
> The target examples in Figure 3 (Section 2) are:
> > “atheism” (Specified target A)
> > “no belief in gods” (Specified target B)
> > “religion” (Specified target C)
> > “believe in god” (Specified target D)
>
> The label sequences in our experimental setting (Figure 4) are (the label order is one instance of label assignment):
> > “Favor, Against, None” (Label Assignment A)
> > “Against, Favor, None” (Label Assignment B)
> > “None, Favor, Against” (Label Assignment C)
>
> Noteworthy, the model prediction variance is **not only related to the above specified instructions**, even small changes to the prompt also make a difference (e.g., change “a tweet” to “the tweet”). We found that model predictions are not consistent when different prompts are provided (even ChatGPT), which poses challenges when machine annotation is applied.
>
> We’ll revise these figures to help the audience better understand.
>
> #### **1.3) For the two proposed methods in Section 3**
> In Section 3, we do both “improving zero-shot approach” as well as “data augmentation” to improve the machine annotation for computational stance detection.
> + To improve the accuracy and reduce the prediction variation in machine annotation, we conducted a “two hop” inference process by adding one instruction for describing the context-object relation.
> + To improve the learning efficiency of the machine annotation, we conducted a diverse sampling with adversarial multi-target labeling (in the last paragraph of Section 3).
>
> #### **1.4) For the follow-up fine-tuning experiment in Section 4**
> **Section 4 is to train a stance detection model using the machine-annotated data generated in Section 3, and evaluate the performance of learning from machine annotation.**
> The reviewer understood the results shown in Table 1 correctly, but we want to emphasize that these experiments were conducted NOT to repeat the machine annotation process, BUT to prove that training on machine-annotated data can produce reasonable results on several benchmark datasets, which highlighted the effectiveness of our proposed methods on machine annotation.
>
> #### **2) For the “learning efficiency” of training on machine-annotated data**
> **The “learning efficiency” is evaluated on the relatively small number of machine-annotated training data.**
> In Section 3, the machine-annotated corpus size of multi-target samples using our proposed methods is 1.2k (see Line 186), whereas the original human-annotated corpus size is 2.9k (2914). In the follow-up stance detection experiments, we fine-tuned the Roberta model that is trained on the 1.2k machine-annotated corpus, and the results on multiple test sets showed its comparable performance to that trained on the 2.9k human-annotated data (see Line 235), clearly demonstrating the promoted “learning efficiency” of our methods.
>
> We will expand on this point in the revised manuscript.
>
> #### **3) For the concern of test data leakage from pre-training models**
> We don’t think the test data leakage will affect the results presented in our work, though it is a general concern.
> + For our machine annotation experiments, Alpaca and Vicuna are Llama-based open-source models [1] trained via general-purpose instruction-tuning, and no stance detection corpora are directly used in their training process. Compared with LLMs that are instruction-tuned on various existing NLP datasets (e.g., FLAN-T5 [2]), our model selection has less concern about the test data leakage issue.
> + **If the models presumably saw the data before, it makes the observed inconsistency issue more pronounced.** This indeed highlights our contribution to the field by bringing this issue and providing one solution to improve the sample quality of machine annotation.
>
> #### **4) For combining data from human and machine annotation**
> In the practical cases, there would be two situations: large-scale human annotation is available or unavailable. As human annotation is time-consuming to scale and cover various domains (most existing stance detection corpora are only labeled on a few stance targets), machine annotation would serve as a bootstrapping step or an alternative. In case people would like to combine human-annotated and machine-annotated data, the quality of machine-annotated data must be on par with the human-annotated ones, otherwise noise will be introduced. In either case, **the quality of machine annotation needs to be taken into account, which is exactly the aim of our work.**
>
>
> #### **5) For the reproducibility of our work**
> We don’t fully understand why the reviewer claims _“I'm quite sure the idea being described here could not be reproduced.”_
> In fact, we will release our code and data for both the machine annotation and stance detection experiments. As we used the open-source models Alpaca and Vicuna for machine annotation, and Roberta for stance detection, we believe that **interested readers will be able to reproduce our experiments following our paper and code.**
> Moreover, the observation of the prediction variation and our methods are not limited to the models and corpora we evaluated in this paper. This work can be readily extended to other models and various data for computational stance detection.
>
> #### **6) For typos, table format, and missing references**
> Thanks for your suggestions on the typos, table format, and missing references, we will revise them accordingly in the manuscript.
>
> ####
> ####
> **We hope the above points address the comments well and the reviewer can appreciate the contributions of our work, and we will greatly appreciate it if the reviewer can re-rate the scores.**
>
> ####
> #### **Reference:**
> [1] Touvron, Hugo, Thibaut Lavril, Gautier Izacard, Xavier Martinet, Marie-Anne Lachaux, Timothée Lacroix, Baptiste Rozière et al. "Llama: Open and efficient foundation language models." arXiv preprint arXiv:2302.13971 (2023).
> [2] Chung, Hyung Won, Le Hou, Shayne Longpre, Barret Zoph, Yi Tay, William Fedus, Eric Li et al. "Scaling instruction-finetuned language models." arXiv preprint arXiv:2210.11416 (2022).

---

### Meta-Review · Area_Chair_AaVV · 2023-09-15

**Recommendation:** 3

**Metareview:**

The paper explores the feasibility of using LLM for data annotation, using the task of computational stance detection as a case study.  The authors find that while LLMs like Alpaca, Vicuna, and GPT-3.5 turbo achieve reasonable performance on a benchmark dataset for stance labeling, models are sensitive to variation along a number of dimensions (such as prompt and label) and show evidence of intrinsic bias.  They then test a multi-target multi-label setup to improve robustness in their annotation task, finding that training on additional machine-labeled examples improves out-of-domain and cross-target performance.

Overall, the paper presents a solid experiment on the use of machine-labeled data.  An initial critique of the paper across reviewers is that the main focus of the paper is unclear in its current state.  Reviewer 5SF3 stated that they did not find that “this paper makes a focused contribution”, citing the fact that it seemed to run a number of experiments on using models to label stance without very much analysis on any given variation in approach.  While it isn’t clear what exact experiments the reviewer found unclearly motivated or analyzed, reviewer jYAZ similarly felt that section 4, which ran the finetuning experiments on RoBERTa, felt disconnected from the rest of the paper, as it suddenly seemed to be evaluating a new model on a wider range of datasets.  While I understood the purpose of section 4 to be testing the actual performance of models trained on data extracted from the multi-target multi-label annotation setup, the presentation of this section is unclear in its current form.  While the paper is a short paper, and thus, has stricter page constraints, ideally the paper would provide more motivation for a. why they are training the RoBERTa models and b. why they are examining performance across a variety of stance datasets, some of which are out-of-domain from the original training set of SemEval-16 Task-6 A.

Another issue with clarity that both reviewers KnpT and jYAZ identified was in the presentation of the results in Figures 2-4, which show the F1 of 3-class stance prediction using 3 different LLMs.  Reviewer KnpT, for example, found it frustrating to have to refer back to the text to understand what the figures were describing.  Reviewer jYAZ found it unclear what was meant by Specified Target A, Instruction Prompt A, … etc.   While the authors clarify in their rebuttal what each of the dimensions on the X-axis of figures 2-4 mean, these should ideally be included in some form in the paper (either in the main body or appendix).

Finally, one issue that may affect the soundness of the paper is the issue of potential test data leakage in the pre-trained models used for annotation.  While the authors argue in their rebuttal that none of the open-source models explicitly include stance-detection corpora in their instruction-tuning process, there is still the potential for data leakage outside of instruction-tuning.  The potential presence of test data in the pretraining data for the LLMs performing annotation does raise concerns over the true robustness and extensibility of using LLM annotation.  The performance improvement of the finetuned-RoBERTa models on out-of-domain test sets, for example, may be attributed to the LLM annotation scheme reproducing the label distribution of data it should not have already seen.  While the authors argue that the instability between the vanilla instruction and the multi-target/multi-label settings provides some evidence against the potential data leakage issue, it’s hard to attribute the instability across different settings directly to the presence of data leakage, as LLMs are notoriously sensitive to how instructions are presented.  While there’s not a feasible approach to directly address the data leakage issue, a more convincing argument that the machine-annotation method is robust to out-of-domain data would be to repeat the section 4 analysis, taking each of the original “out-of-domain” datasets as the basis of their training corpora once and testing every other dataset against the model trained on the “leave-one-out” dataset.

---

### Decision · Program_Chairs · 2023-10-07

**Decision:**

Accept-Findings

**Comment:**

The paper explores the feasibility of using LLM for data annotation, using the task of computational stance detection as a case study.  The authors find that while LLMs like Alpaca, Vicuna, and GPT-3.5 turbo achieve reasonable performance on a benchmark dataset for stance labeling, models are sensitive to variation along a number of dimensions (such as prompt and label) and show evidence of intrinsic bias.  They then test a multi-target multi-label setup to improve robustness in their annotation task, finding that training on additional machine-labeled examples improves out-of-domain and cross-target performance.

Overall, the paper presents a solid experiment on the use of machine-labeled data.  An initial critique of the paper across reviewers is that the main focus of the paper is unclear in its current state.  Reviewer 5SF3 stated that they did not find that “this paper makes a focused contribution”, citing the fact that it seemed to run a number of experiments on using models to label stance without very much analysis on any given variation in approach.  While it isn’t clear what exact experiments the reviewer found unclearly motivated or analyzed, reviewer jYAZ similarly felt that section 4, which ran the finetuning experiments on RoBERTa, felt disconnected from the rest of the paper, as it suddenly seemed to be evaluating a new model on a wider range of datasets.  While I understood the purpose of section 4 to be testing the actual performance of models trained on data extracted from the multi-target multi-label annotation setup, the presentation of this section is unclear in its current form.  While the paper is a short paper, and thus, has stricter page constraints, ideally the paper would provide more motivation for a. why they are training the RoBERTa models and b. why they are examining performance across a variety of stance datasets, some of which are out-of-domain from the original training set of SemEval-16 Task-6 A.

Another issue with clarity that both reviewers KnpT and jYAZ identified was in the presentation of the results in Figures 2-4, which show the F1 of 3-class stance prediction using 3 different LLMs.  Reviewer KnpT, for example, found it frustrating to have to refer back to the text to understand what the figures were describing.  Reviewer jYAZ found it unclear what was meant by Specified Target A, Instruction Prompt A, … etc.   While the authors clarify in their rebuttal what each of the dimensions on the X-axis of figures 2-4 mean, these should ideally be included in some form in the paper (either in the main body or appendix).

Finally, one issue that may affect the soundness of the paper is the issue of potential test data leakage in the pre-trained models used for annotation.  While the authors argue in their rebuttal that none of the open-source models explicitly include stance-detection corpora in their instruction-tuning process, there is still the potential for data leakage outside of instruction-tuning.  The potential presence of test data in the pretraining data for the LLMs performing annotation does raise concerns over the true robustness and extensibility of using LLM annotation.  The performance improvement of the finetuned-RoBERTa models on out-of-domain test sets, for example, may be attributed to the LLM annotation scheme reproducing the label distribution of data it should not have already seen.  While the authors argue that the instability between the vanilla instruction and the multi-target/multi-label settings provides some evidence against the potential data leakage issue, it’s hard to attribute the instability across different settings directly to the presence of data leakage, as LLMs are notoriously sensitive to how instructions are presented.  While there’s not a feasible approach to directly address the data leakage issue, a more convincing argument that the machine-annotation method is robust to out-of-domain data would be to repeat the section 4 analysis, taking each of the original “out-of-domain” datasets as the basis of their training corpora once and testing every other dataset against the model trained on the “leave-one-out” dataset.